# Dapagliflozin Impact on the Exercise Capacity of Non-Diabetic Heart Failure with Reduced Ejection Fraction Patients

**DOI:** 10.3390/jcm11102935

**Published:** 2022-05-23

**Authors:** João Reis, Ana Rita Teixeira, António Valentim Gonçalves, Rita Ilhão Moreira, Tiago Pereira Silva, Ana Teresa Timóteo, Rui Cruz Ferreira

**Affiliations:** Department of Cardiology, Hospital de Santa Marta, Centro Hospitalar e Universitário de Lisboa Central, Rua de Santa Marta 50, 1169-024 Lisbon, Portugal; rita.ribeiro.teixeira@gmail.com (A.R.T.); antonio.a.goncalves.14@gmail.com (A.V.G.); ritailhaomoreira@gmail.com (R.I.M.); tiagopsilva@sapo.pt (T.P.S.); ana_timoteo@yahoo.com (A.T.T.); cruzferreira@netcabo.pt (R.C.F.)

**Keywords:** heart failure, Heart Failure with reduced Ejection Fraction, cardiopulmonary exercise test, peak oxygen uptake, sodium–glucose co-transporter 2 inhibitors

## Abstract

Background: Dapagliflozin has been shown to reduce morbidity and mortality in Heart Failure with reduced Ejection Fraction (HFrEF), but its impact on exercise capacity of non-diabetic HF outpatients is unknown. Methods: Adult non-diabetic HF patients with a left ventricular ejection fraction (LVEF) <50% were randomized 1:1 to receive dapagliflozin 10 mg or to continue with HF medication. Patients underwent an initial evaluation which was repeated after 6 months. The variation of several clinical parameters was compared, with the primary endpoint being the 6 month peak oxygen uptake (pVO_2_) variation. Results: A total of 40 patients were included (mean age 61 ± 13 years, 82.5% male, mean LVEF 34 ± 5%), half being randomized to dapagliflozin, with no significant baseline differences between groups. The reported drug compliance was 100%, with no major safety events. No statistically significant difference in HF events was found (*p* = 0.609). There was a 24% reduction in the number of patients in New York Heart Association (NYHA) class III in the treatment group as opposed to a 15.8% increase in the control group (*p* = 0.004). Patients under dapagliflozin had a greater improvement in pVO_2_ (3.1 vs. 0.1 mL/kg/min, *p* = 0.030) and a greater reduction in NT-proBNP levels (−217.6 vs. 650.3 pg/mL, *p* = 0.007). Conclusion: Dapagliflozin was associated with a significant improvement in cardiopulmonary fitness at 6 months follow-up in non-diabetic HFrEF patients.

## 1. Background

Dapagliflozin is a drug from the renal sodium–glucose cotransporter 2 inhibitor (SGLT2i) class, initially approved for therapy in patients with type 2 diabetes mellitus (DM) [1]. Several studies of this drug class have shown an important reduction in the incidence of cardiovascular events and hospitalizations for heart failure (HF) in patients with DM type 2 who had or were at risk for atherosclerotic cardiovascular disease [2,3,4]. The EMPA-REG OUTCOME trial showed a 35% reduction in the relative risk of hospitalization for HF in patients with DM type 2 with the use of Empagliflozin [2]. This was followed by the CANVAS [3] and DECLARE-TIMI 58 [4] studies, demonstrating a 33% and 27% reduction with the use of Canagliflozin and Dapagliflozin, respectively. This raised the question of the potential benefit of this drug class in patients with HF.

A multicenter, randomized study showed a significant reduction in the incidence of hospitalizations for HF, cardiovascular mortality, and total mortality in patients with Heart Failure with reduced Ejection Fraction (HFrEF), regardless of being diabetic—DAPA-HF [5]. These results were corroborated by the EMPEROR-Reduced study, which revealed a class effect of SGLT2i in the reduction in cardiovascular mortality/HF hospitalization [6]. Addition of SGLT2i to triple neurohormonal blockade therapy was associated with an estimated 13% reduction in overall mortality risk and a 14% reduction in CV mortality risk, accompanied by a 26% reduction in the combined relative risk of 26% CV mortality/first hospitalization for HF and 25% risk of CV mortality/recurrent hospitalization for HF [7]. This effect was consistent across different subgroups, namely New York Heart Association (NYHA) functional class, ejection fraction, renal function, presence of diabetes and baseline HF risk models [8].

In this context, SGLT2i has recently been approved as the fourth pillar of prognostic-modifying therapy for HFrEF, added to the gold–standard treatment of the triple neurohormonal blockade strategy [9,10], with this drug combination being expected to prolong the life of a 70-year-old HFrEF patient around 5 years compared with placebo [11].

The mechanisms underlying the reduction in HF events are not yet fully clarified and are most likely multifactorial [12], with some evidence suggesting some degree of positive left ventricular remodeling [13,14,15,16,17]. Despite the prognostic benefit of this drug class, its impact on functional capacity is largely unknown with recent trials having divergent results: on the Emperial trials, empagliflozin failed to increase the primary endpoint of change in 6 min walk distance compared with placebo on patients with HF with reduced and preserved ejection fraction [18], while, on EMPA-TROPISM, it was associated with a significant improvement in exercise capacity assessed by peak oxygen uptake and 6 min walk distance [14]. There is limited evidence on the impact of SGLT2i on cardiopulmonary exercise test (CPET) parameters, which is the gold-standard to evaluate functional capacity in HF, being a powerful prognostic predictor and a useful discriminative tool to guide patient referral for advanced HF interventions [19]. Our study aimed to assess the safety and efficacy of the drug in non-diabetic patients with HFrEF and to evaluate its impact on their functional capacity, assessed by cardiopulmonary exercise testing, and left ventricular remodeling.

## 2. Methods

### 2.1. Patient Population and Study Design

This is a single-site open-label prospective randomized study including non-diabetic patients with HF with reduced left ventricular ejection fraction followed in our institution by the HF team. Randomization was performed 1:1 using an online randomization platform and patients were randomized according to gender, LVEF and HF etiology. Patients were randomly assigned to take Dapagliflozin 10 mg orally once daily, in addition to recommended HF therapy, or to maintain their usual medication for a period of 6 months. Being an open-label study, both the health providers and the patients were aware of the treatment being given.

Patients were considered eligible if they met the following criteria: Heart Failure with reduced Ejection Fraction (left ventricular ejection fraction less than 50%); NYHA functional class ≥ II; medical treatment previously optimized according to the current guidelines for at least 3 months; patients treated according to current guidelines for coronary artery disease, valvular disease, atrial fibrillation and with cardiac implantable electronic devices (CIED) as indicated; and ability to safely preform a maximal cardiopulmonary exercise test (CPET).

Exclusion criteria included previous history of DM (namely, HbA1c ≥ 6.5%) (*n* = 38); glomerular filtration rate <30 mL/min according to the Cockrauft-Gault formula (*n* = 29); symptoms of hypotension or systolic blood pressure below 90 mmHg (*n* = 21); cardiac procedures planned for the next 6 months (either revascularization procedures, cardiac surgery or CIED implantation) (*n* = 18); age under 18 (*n* = 4); pregnant woman or woman with a desire to become pregnant (*n* = 2); urinary tract infection in the last month (*n* = 6); and medical treatment up-titration during the previous 3 months (*n* = 48). Patients who underwent any relevant therapeutic intervention during follow-up were excluded from the analysis.

### 2.2. Baseline and Follow-Up Evaluation

At baseline, each patient underwent a clinical evaluation by the attending physician at the outpatient clinic, including clinical, laboratorial, electrocardiographic, echocardiographic and cardiopulmonary exercise testing (CPET) data, which was repeated at 6 months follow-up. Data were collected during the clinical evaluation in an outpatient setting.

Guideline-directed medical therapy and overall patient management was left at the discretion of the assistant cardiologist.

The following parameters were specifically recorded:Anthropometric and physical examination data: age, gender, weight, height, body mass index, heart rate and blood pressure.Clinical data: etiology of HF, assessment of control of the patient’s comorbidities and record of the patient’s medication; NYHA functional class and risk assessment through risk scores validated in HF (Heart Failure Survival Score, Seattle Heart Failure Model and Heart Failure Risk Calculator).Functional capacity data: peak oxygen consumption (pVO_2_), ventilation/CO_2_ production slope (VE/VCO_2_ slope), time to anaerobic threshold and heart rate recovery after the first minute of effort.Echocardiography parameters: left ventricular dimension, left ventricular ejection fraction, left ventricular strain, right ventricular function, RV–AD gradient, valvular function, and other summary examination parameters.Blood tests, including hemoglobin, creatinine, high-sensitivity troponin I and NTproBNP levels, glycated hemoglobin and lipid profile.Record of concomitant therapy with a potential impact on the evolution of HF, such as the use of intravenous iron therapy, implantation of CIED, changes of HF, surgical or percutaneous valve procedures, coronary procedures, and arrhythmia ablation procedures.The occurrence of major cardiovascular events was also recorded, including death (all causes, cardiovascular and due to HF), hospitalizations (from cardiovascular causes and due to HF) and need for advanced HF therapy (cardiac transplantation and left ventricular assist device implantation).

#### Cardiopulmonary Exercise Testing

A maximal symptom-limited treadmill CPET, defined by peak respiratory exchange rate (RER) >1.05, was performed using the modified Bruce protocol (GE Marquette Series 2000 treadmill). Gas analysis was preceded by calibration of the equipment. Minute ventilation, oxygen uptake and carbon dioxide production were acquired breath-by-breath, using a SensorMedics Vmax 229 gas analyzer. The pVO_2_ was defined as the highest 30 s average achieved during exercise and was normalized for body mass. The anaerobic threshold was determined by combining the standard methods (V-slope preferentially and ventilatory equivalents). The VE/VCO_2_ slope was calculated by least squares linear regression, using data acquired throughout the whole exercise. COP was measured as the minimum value of the ventilatory equivalent for oxygen (VE/VO_2_ minimum). Partial pressure of end-tidal carbon dioxide (P_ET_CO_2_) was reported before exercise (P_ET_CO_2AR_), at anaerobic threshold (P_ET_CO_2AT_) and at peak exercise in mmHg units, and the increase during exercise until the anaerobic threshold is achieved (P_ET_CO_2DIF_) was also calculated. Peak oxygen pulse (PP) was calculated by dividing derived pVO_2_ by the maximum heart rate (HR) during exercise and was expressed in milliliters per beat. Circulatory power was calculated as the product of pVO_2_ and peak systolic blood pressure and the ventilatory power was calculated by dividing peak systolic blood pressure (BP) by the VE/VCO_2_ slope. Several composite parameters of CPET were also automatically calculated.

### 2.3. Follow-Up and Endpoint

All patients were followed up for 6 months from the date of inclusion and baseline evaluation. The variation of each clinical, laboratorial, echocardiographic and CPET parameter between baseline and 6 months follow-up evaluation was compared between groups, as was the occurrence of mortality and HF events. The primary endpoint of this study was the difference in pVO_2_ 6 month variation between groups. LV reverse remodeling was assessed as a secondary endpoint, namely the difference in LV end-diastolic diameter 6 month variation, as well as was the NT-proBNP variation.

### 2.4. Ethics

This investigation follows the principles outlined by the Declaration of Helsinki. The institutional ethics committee approved the study protocol. All patients provided written informed consent. Participation was entirely voluntary, and the participants could withdraw at any time or refuse to provide part or all the data. All information collected remained confidential. Only researchers and the included patients will have access to the data. The anonymity of patients was guaranteed, and each patient was assigned a code, and the code–patient relationship will only be owned by the investigators.

### 2.5. Statistical Analysis

We estimated that a sample of 16 individuals in each arm would be required so that an increase in peak oxygen uptake of 2 mL/kg/min would be statistically significant, with 95% confidence and 80% potency [20]. In order to compensate for eventual losses during follow-up, the predicted sample size was increased by 25% (total of 40 patients).

Baseline characteristics and follow-up workup results were summarized as frequencies (percentages) for categorical variables, as means and standard deviations for continuous variables when normality was verified and as median and interquartile range when normality was not verified by the Kolmogorov–Smirnov test. Student’s *t*-test for independent samples or the Mann–Whitney test (when normality was not confirmed) was used for all comparisons. The Chi-Square test or Fisher’s exact was were used to compare categorical variables.

Repeated-measures ANOVA with time as a within-subjects effect and treatment with dapagliflozin as between subjects effect were implemented to assess the effect of time (baseline vs. 6 months), treatment (yes/no) and their interaction regarding continuous outcomes. Partial eta2 was calculated for assessing effect size considering >0.2 (small), >0.5 (moderate) and >0.8 (large). Generalized estimating equations were implemented to assess NYHA III class (no vs. yes) change across time, groups of treatment with dapagliflozin and time × treatment interaction. Effect size was assessed with odds ratios.

Treatment effect was assessed by comparing the mean variation of each continuous variable between the two groups (variation was calculated by subtracting the follow up score from the baseline score). A two-tailed probability value of <0.05 was considered statistically significant.

Data were analyzed using the software Statistical Package for the Social Science for Windows, version 24.0 (SPSS Inc., Chicago, IL, USA).

## 3. Results

A total of 40 patients were enrolled after application of exclusion criteria, of which 20 patients were randomly allocated to the Dapagliflozin group and 20 patients to the conventional treatment group (control group), according to age, HF etiology and LVEF. Six month follow-up data were available for all patients in both groups (Figure 1).

Detailed baseline population characteristics are described in Table 1. The mean age of the study cohort was 60.9 ± 13.0 years, and 82.5% were men. More than three-quarters of the study population (78.0%) had an ischemic etiology of HF and 27.5% had at least one HF hospitalization in the previous 12 months, with a mean LVEF of 34.1 ± 8.3% and a median baseline NT-proBNP value of 781.0 (350.7–1599.1) pg/mL. One-fifth of the population was in NYHA functional class III.

The demographic and clinical profiles of the two groups were well balanced (Table 1). The symptomatic burden, HF clinical risk scores and traditional HF prognostic markers, such as LVEF, NT-proBNP values and peak oxygen uptake were also similar between the study arms, as well as atrial fibrillation prevalence. There were no differences regarding neurohormonal blockade therapy or CIED implantation rates. However, patients randomized to receive dapagliflozin had a higher pulmonary artery systolic pressure (*p* = 0.040), whereas patients in the control group had a higher baseline high-sensitivity troponin I value (*p* = 0.041).

Overall, 15.0% of patients in the study population experienced major adverse cardiovascular events (MACEs) during the first 6 month follow-up; however, no deaths nor heart transplant/ left ventricular assistance device implantations were reported (Table 2). There were no significant differences in the individual MACE components between the groups.

Additionally, there were no major drug-related adverse events, with no patients having to discontinue the drug. There were no cases of symptomatic hypotension requiring drug adjustments, hypoglycemia (blood glucose levels below 70 mg/dL) nor any cases of diabetic ketoacidosis, with one reported urinary tract infection in a male patient in the SGLT2i arm.

Dapagliflozin use led to an increase in the primary endpoint, with a more significant increase in pVO_2_ in this group in comparison to the control group: 3.1 vs. 0.1 mL/kg/min, *p* = 0.027 (Figure 2). This was accompanied by a significant improvement in other cardiopulmonary fitness, such as VE/VCO_2_ (−0.8 vs. 3.3, *p* = 0.027)—Figure 3.

At 6 months follow-up, there was statistically significantly variation in the proportion of patients in NYHA class III between groups, with a decrease in the patients in the dapagliflozin group from 28.6% to 4.8% and an increase in the control group from 10.5% to 26.3% (*p* = 0.041)—Table 3. Patients in the intervention group also had an improvement in their risk profile, as assessed by a significant increase in the SHFM-estimated 1 year survival rate (89.5 to 92.5 vs. 88.3 vs. 87.6, *p* = 0.014) and by a numerical improvement in the MAGGIC score, which did not attain statistical significance (*p* = 0.073). However, there was no difference between groups regarding the variation of the HFSS. Concomitantly, patients under dapagliflozin experienced a significant decrease in natriuretic peptides during follow-up as opposed to the control group, from 1201.5 to 983.9 pg/mL and from 1132.1 to 1782.4 pg/mL (*p* = 0.007), respectively, despite no difference on loop diuretic dose or other HF drugs.

As expected, patients under iSGLT2 had a significant reduction in the glycated hemoglobin during follow-up (5.8 to 5.6% vs. 5.7 to 6.1%, *p* < 0.001), while there was no significant effect of the drug on the renal function trajectory: glomerular filtration rate variation −1.9 ± 1.6 vs. −6.6 ± 0.6, *p* = 0.144.

During follow-up, patients in the intervention arm experienced a significant remodeling with a significant decrease in the left-ventricular end-diastolic diameter (LVEDD), from 67.2 to 63.0 mm vs. from 65.1 to 65.8 mm in the control group (*p* = 0.010), while there was no significant difference in the left-ventricular end-systolic diameter (LVESD) variation (*p* = 0.460)—Figure 4. The dapagliflozin arm also had an improvement in global longitudinal strain from −8.2 ± 3.2 to −10.4 ± 2.3 as opposed to the control group (from −8.4 ± 2.8 to −6.8 ± 2.8, *p* < 0.001), while there was no significant difference in LVEF or right ventricular systolic function assessed by TAPSE (*p* = 0.512 and *p* = 0.376, respectively). There was also a significant improvement in LV-filling pressures parameters among patients in the active arm.

## 4. Discussion

In recent decades, evidence-based pharmacological and device therapies led to an important event rate reduction in HF patients. However, a substantial proportion of patients remains at a high risk for hospitalization and is estimated that 1–10% of the overall HF population will progress to an advanced stage of the disease. HF hospitalizations have a dramatic impact on patients’ quality of life and are a strong predictor of mortality, whose risk increases significantly with each hospitalization, with an 1 year all-cause mortality of 23.6% [21,22].

The finding that SGLT2i reduced the risk of a first hospitalization for HF in patients with type 2 diabetes raised a new perspective for the care of HF patients [2,3,4]. Accumulating evidence from other RCTs, whose results were published after this study’s design, established a class effect on HF for SLT2i [8]. Furthermore, with the published results of EMPEROR-Preserved and SOLOIST-WHF trials, there is a growing body of evidence suggesting a beneficial effect of SGLT2i on heart failure events across the spectrum of LVEF [23,24,25]. The pleiotropic mechanisms underlying the beneficial CV effects of SGLT-2is in patients with HF are not fully understood, but cannot be explained exclusively by their diuretic or glucose-lowering effects because its modest hypoglycemic activity is comparable to other glucose-lowering drugs [12].

This analysis intended to demonstrate the real-life safety and beneficial effect of SGLT2i use in non-diabetic patients with HF with reduced LVEF, based on a multiparametric evaluation of several HF prognostic markers. There are some key differences between our study’s population baseline profile and that of the DAPA-HF trial, as our population was significantly younger and it did not include diabetic patients, with a higher proportion of male patients than in DAPA-HF5. Furthermore, our study included patients with less severe disease, namely HF with mildly reduced EF5, while DAPA-HF enrolled patients with a LVEF 40% or less [5]. In this regard, the mean LVEF is higher than that of the aforementioned RCT, the mean baseline NT-proBNP value lower and there was a lower proportion of patients in functional class NYHA III or with HF hospitalizations in the previous 12 months. Additionally, there was a greater proportion of patients with ischemic HF, a greater use of ARNI and ICD implantation rates. Unlike in the published RCTs [5,6], SGLT2i use was not associated with a lower rate of HF events or mortality (*p* = 0.720), which may have been related to the overall rate of MACE and to the short follow-up period (mean follow-up time of 6 months as opposed to 18.2 months in DAPA-HF). No patients had to stop Dapagliflozin and there were no serious adverse events, while 10.5% of patients under dapagliflozin in DAPA-HF interrupted the drug, with 1.2% having serious adverse events related to volume depletion and 1.6% serious renal adverse events [5].

Patients under SGLT2 experienced a symptomatic improvement, with a 23.8% reduction in patients in NYHA functional class III at 6 month follow-up, while patients in the control group had a symptomatic deterioration, which is in line with both EMPA-TROPISM [14] and DEFINE-HF [26] that revealed a quality-of-life improvement assessed by the Kansas City Cardiomyopathy Questionnaire. This was accompanied by a significant improvement in the primary endpoint, with a 3.1 mL/kg/min increase in pVO_2_ on patients under treatment as compared to a minimal increase of 0.1 mL/kg/min on the control group (*p* = 0.030) and an improvement in VE/VCO_2_, which is a submaximal exercise variable. CPET provides information on the functional capacity and outcome prediction in HF, with pVO_2_ being the mostly useful parameter to guide heart transplantation referral and interventions that lead to an increase in pVO_2_ are associated with a prognostic benefit. In EMPA-TROPISM [14], the pVO_2_ was significantly increased in nondiabetic HFrEF treated patients by 1.1 mL/kg/min versus a 0.5 mL/kg/min decline in the placebo, which is in line with the functional capacity improvement in our study. Furthermore, there was also a non-significant trend toward improvement in the VE/VCO_2_ slope in the empagliflozin group, whereas dapagliflozin led to a statistically significant improvement in our population. This contrasts with the neutral results of the EMPERIAL trials regarding the use of empagliflozin on 6 min walking test distance [18]. However, the choice of 6 min walking test as a primary endpoint may be the reason for its neutral results, as it is not an optimal measure of improvement in patients’ HF status, being affected by many HF-related comorbidities and lacking reproducibility. Furthermore, it is not usually used as a primary endpoint in HF trials, since CPET provides a much more refined assessment of cardiopulmonary fitness and is a more sensitive parameter of exercise capacity than 6MWT [18,19]. The results from the DETERMINE trials may shed more light on this matter.

Natriuretic peptide concentration variation after treatment has prognostic significance as treatments leading to a greater relative reduction are associated with a better prognosis and discharge natriuretic peptide concentrations are an excellent predictor of 1 year death or re-hospitalization among patients with acute HF [27]. Patients under dapagliflozin had a significant reduction from baseline to 8 months in NT-proBNP values (*p* = 0.007); however, the effect of SGTL2i on natriuretic peptides is not well understood. Our results are in line with those of DAPA-HF (−196 ± 2387 vs. 101 ± 2944, *p* < 0.001) [5] as opposed to the DEFINE-HF trial, during which the use of dapagliflozin over 12 weeks did not affect mean NT-proBNP, despite increasing the proportion of patients experiencing clinically meaningful improvements in HF-related health status [26].

Despite the available evidence of the protective effect of this class of drugs on the decline of renal function and reduction in renal events, during 6 months follow-up, there was no statistically significant difference on the variation of the estimated GFR between groups, which may be explained by the more prolonged follow-up period of those RCTs (median follow-up of 2.4 years in DAPA-CKD) [28].

Several studies evaluated the effect of SGLT2i use on several echocardiographic parameters, including systolic and diastolic function markers, both in diabetic patients with evidence of cardiovascular disease and HF patients [13,14,15,16,17]. The EMPA-HEART study reported LV mass regression on diabetic patients without HF [16] and EMPA-TROPISM expanded the drug’s positive impact on LV remodeling to non-diabetic HFrEF patients, with a significant reduction in LV volumes and improvements in LVEF in the empagliflozin-treated patients [14], which contrasts with the REFORM trial that did not find any improvement in LV remodeling with dapagliflozin on diabetic HF patients [17]. In our study, patients in the intervention group also experienced a positive LV remodeling, as assessed by LVEDD and LVESD, and an improvement in LV longitudinal myocardial strain, which is a more refined marker of systolic function than LVEF [15]. GLS adds incremental value to LVEF in the prediction of adverse outcomes and can play a significant role in improving risk stratification in HF. In line with a previously published study, dapagliflozin treatment led to a significant improvement in LV diastolic dysfunction and a significant decrease in estimated LV filling pressure [29]. Since LV diastolic function has been acknowledged as a determinant of symptoms and prognosis in HF, these data reinforce the role of SGLT2i as novel agents for HF treatment and may suggest a potential benefit on HFpEF where diastolic dysfunction plays a major role.

Despite DAPA-HF not having included patients with LVEF >40%, our analysis suggests a potential beneficial effect on functional capacity and LV reverse remodeling in LVEF up to 50% (including the spectrum of HF with mid-range or mildly reduced ejection fraction). This is in line with recent evidence that showed the magnitude of the effect of empagliflozin on HF outcomes was clinically meaningful and similar in patients with LVEF <25% to 65% [23,24,25].

Our work reinforces the extremely favorable safety profile of Dapagliflozin, with 100% drug compliance at 6 months follow-up and no major adverse effects, being one of the first trials to assess the impact of SGLT2i on CPET parameters of HFrEF patients. Its positive impact on exercise fitness, the reduction in NT-proBNP levels and positive LV remodeling, may translate into a reduction in HF events, leading to a significant reduction in HF hospitalizations during a longer follow-up.

### Study Limitations

There are limitations in our study that should be mentioned, including its single-centre design and the small sample size, which may affect the validity and reproducibility of our work’s conclusions. Despite having performed CPET to evaluate cardiopulmonary fitness, functional capacity assessment through NYHA functional class can be subjective, and it is important to assess the effect on both symptomatic status and QoL, using a more refined tool, such as the Kansas City Cardiomyopathy Questionnaire. Moreover, this study’s population is very heterogeneous, including advanced HF patients on a waiting list of advanced therapies and patients with less severe heart disease, with LVEF ranging from severely reduced to mid-range. Additionally, our study’s conclusions cannot be extended to patients with diabetes melitus and chronic kidney disease stage 4 and 5, as these patients have been excluded from the analysis. Furthermore, there was also a selection bias, as patients unable to perform a maximal CPET were excluded. Finally, due to the rather short follow-up period, the reduced population size and the low rate of events, it was not possible to ascertain the impact of dapagliflozin use in MACE and HF events in our analysis and to confirm if its use in a real-world setting is associated with the same benefits on hard endpoints as DAPA-HF revealed.

## 5. Conclusions

In this single-centre experience, Dapagliflozin use in non-diabetic HF patients with LVEF <50% proved to be a safe strategy and led to a significant improvement in natriuretic peptide levels, exercise performance and echocardiographic markers of LV systolic function at 6 months follow-up. It is one of the first studies to demonstrate a pVO_2_ increase, which may explain the drug’s positive impact on HFrEF patients’ long-term prognosis. The results of this trial of Dapagliflozin further establish the pivotal role of iSLT2 in the management of HF patients.

## Figures and Tables

**Figure 1 jcm-11-02935-f001:**
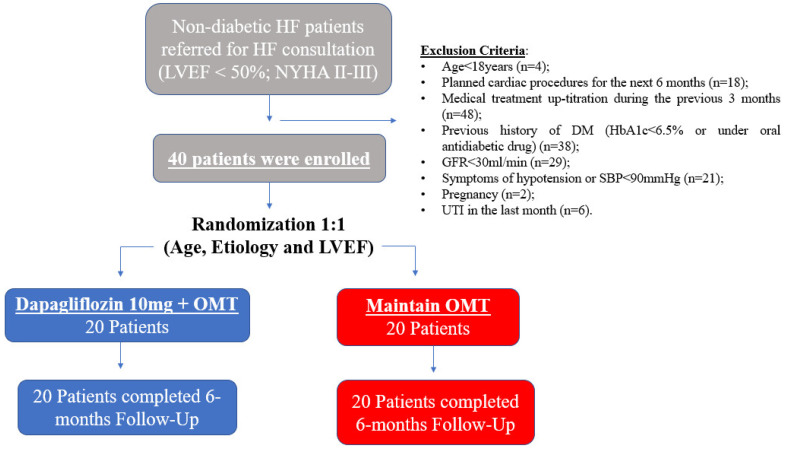
Study design and population selection.

**Figure 2 jcm-11-02935-f002:**
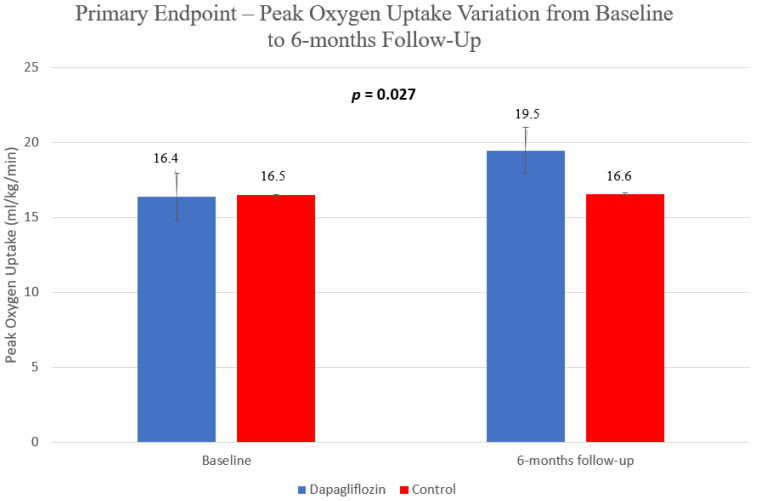
Primary endpoint—peak oxygen uptake variation from baseline to 6 months follow-up.

**Figure 3 jcm-11-02935-f003:**
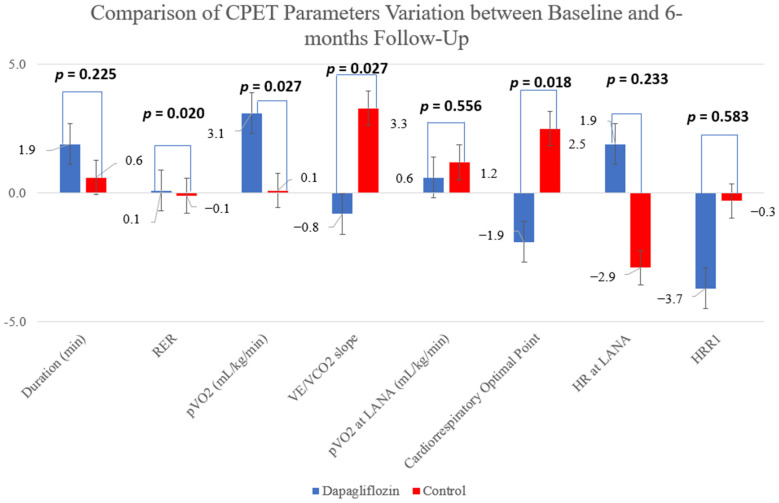
Comparison of CPET parameters variation between baseline and 6 month follow-up.

**Figure 4 jcm-11-02935-f004:**
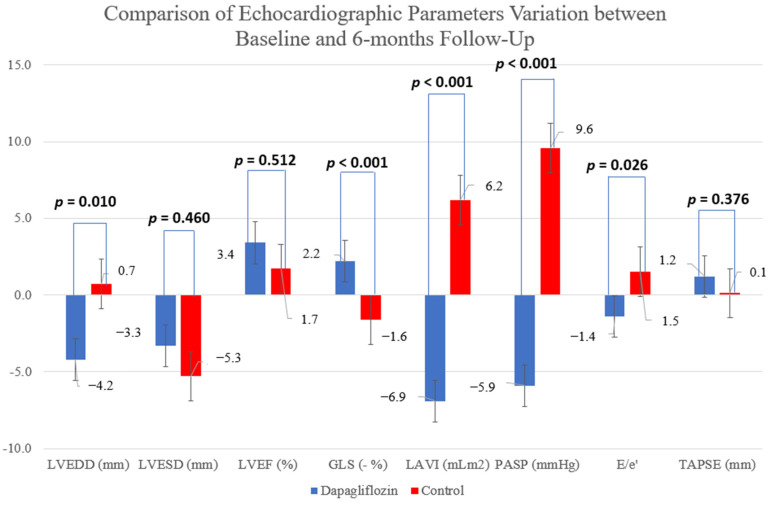
Comparison of echocardiographic parameters variation between baseline and 6 months follow-up.

**Table 1 jcm-11-02935-t001:** Comparison of the baseline clinical characteristics between the two groups.

	Total Population (*n* = 40)	Dapagliflozin (*n* = 20)	Control (*n* = 20)	*p* Value
Age (years)	60.9 ± 13.0	60.3 ± 11.6	61.7 ± 14.8	0.740
Male gender	33 (82.5%)	17 (85.0%)	16 (80.0%)	0.787
Ischemic etiology	28 (78.0%)	16 (80.0%)	12 (60.0%)	0.369
Previous MI	28 (78.0%)	16 (80.0%)	12 (60.0%)	0.369
Previous PCI	28 (78.0%)	16 (80.0%)	12 (60.0%)	0.369
Previous CABG	3 (7.5%)	3 (15.0%)	0 (0%)	0.087
Previous valvular heart surgery	5 (12.5%)	3 (15.0%)	2 (10.0%)	0.720
HF hospitalizations in the previous 12 months	11 (27.5%)	8 (40.0%)	3 (15.0%)	0.115
Heart failure medication				
beta-blocker	39 (97.5%)	20 (100%)	19 (95.0%)	0.675
ACE-i/ARB/ARNI	40 (100%)	20 (100%)	20 (100%)	1.000
MRA	36 (90.0%)	19 (95.0%)	17 (85.0%)	0.510
Ivabradine	7 (17.5%)	3 (15.0%)	4 (20.0%)	0.675
Digitalis	2 (5.0%)	0 (0%)	2 (10.0%)	0.227
DAPT	6 (15.0%)	2 (10.0%)	4 (20.0%)	0.308
Oral anticoagulation	19 (47.5%)	8 (40.0%)	11 (55.0%)	0.210
Furosemide dose (mg)	30.5 ± 31.0	25.7 ± 29.8	33.7 ± 32.6	0.421
Ferric carboxymaltose	17 (42.5%)	13 (65.0%)	4 (20.0%)	**0.012**
Cardiac rehabilitation	21 (52.5%)	12 (60.0%)	9 (45.0%)	0.563
Hypertension	25 (62.5%)	14 (70.0%)	11 (55.0%)	0.567
Dyslipidemia	26 (65.0%)	16 (80.0%)	10 (50.0%)	0.119
Current or former smoker	28 (70.0%)	17 (85.0%)	11 (55.0%)	0.334
Atrial fibrillation	12 (30.0%)	6 (30.0%)	6 (30.0%)	1.000
ICD	25 (62.5%)	14 (70.0%)	11 (55.0%)	0.567
CRT device	6 (15.0%)	3 (15.0%)	3 (15.0%)	1.000
Chronic kidney disease	8 (20.0%)	5 (25.0%)	3 (15.0%)	0.698
Peripheral artery disease	9 (22.5%)	4 (20.0%)	5 (25.0%)	0.712
COPD	11 (27.5%)	5 (25.0%)	6 (30.0%)	0.583
NYHA Class III–IV	8 (20.0%)	6 (30.0%)	2 (10.0%)	0.241
HFSS	8.6 ± 0.7	8.7 ± 0.6	8.6 ± 0.7	0.871
SHFM	88.9 ± 5.9	89.5 ± 6.7	88.3 ± 4.9	0.517
MAGGIC score	17.9 ± 5.4	18.1 ± 5.4	17.8 ± 5.6	0.860
Hemoglobin (g/dL)	13.9 ± 1.4	14.0 ± 1.6	13.8 ± 1.2	0.707
Creatinine (mg/dL)	1.1 ± 0.3	1.2 ± 0.3	1.1 ± 0.3	0.495
GFR (mL/min)	70.4 ± 20.6	68.7 ± 23.8	72.5 ± 17.1	0.596
hs-cTnI (ng/mL)	15.2 ± 31.6	9.2 ± 6.7	21.9 ± 45.1	**0.041**
NT-proBNP (pg/mL)	781.0 (350.7–1599.1)	890.5 (426.5–1652.0)	747.4 (287.7–1490.2)	0.881
HbA1c (%)	5.8 ± 0.4	5.8 ± 0.4	5.7 ± 0.5	0.352
LDL (mg/dL)	90.3 ± 35.0	85.2 ± 33.2	95.8 ± 36.9	0.150
LVEDD (mm)	66.1 ± 7.2	65.1 ± 6.0	67.2 ± 8.4	0.373
LVESD (mm)	48.3 ± 11.9	49.4 ± 9.6	46.9 ± 14.3	0.522
LVEF (%)	34.1 ± 8.3	34.5 ± 8.9	33.5 ± 7.8	0.708
GLS (%)	8.2 ± 3.0	8.2 ± 3.2	8.4 ± 2.8	0.866
TAPSE (mm)	18.8 ± 4.9	19.0 ± 5.7	18.6 ± 4.2	0.831
LA indexed volume (mL/m^2^)	30.7 ± 14.4	43.4 ± 12.2	41.9 ± 17.5	0.740
PASP (mmHg)	33.0 ± 9.8	36.6 ± 7.8	29.8 ± 10.4	**0.040**
E/e’	12.5 ± 4.7	13.3 ± 4.5	11.8 ± 3.1	0.372
CPET duration (min)	10.4 ± 3.6	10.0 ± 4.0	10.9 ± 3.2	0.451
Peak RER	1.10 ± 0.11	1.11 ± 0.12	1.09 ± 0.12	0.661
pVO_2_ (mL/kg/min)	16.5 ± 4.5	16.4 ± 3.9	16.5 ± 5.1	0.921
VE/VCO_2_	34.3 ± 8.3	34.4 ± 7.9	34.1 ± 8.9	0.919

Values are the mean ± SD; NT-proBNP values are expressed as median (interquartile range). Bold values denote statistical significance at the *p* < 0.05 level. ACE-I—Angiotensin-converting Enzyme Inhibitors; ARB—Angiotensin II Receptor Blocker; ARNI—Angiotensin Receptor-Neprilysin Inhibitor; CABG—Coronary Artery Bypass Grafting; COPD—Chronic Obstructive Pulmonary Disease; CRT—Cardiac Resynchronization Therapy; DAPT—Dual Antiplatelet Therapy; HF—Heart Failure; ICD—Implantable Cardioverter-Defibrillator; iSGLT2—Sodium–Glucose Co-Transporter 2 Inhibitors; MRA—Mineralocorticoid Receptor Antagonist; MI—Myocardial Infarction; PCI—Percutaneous Coronary Intervention. CPET—Cardiopulmonary Exercise Testing; GLS—Global Longitudinal Strain; HbA1c—Glycated Hemoglobin; HFSS—Heart Failure Survival Score; hs-cTnI—High-Sensitivity Troponin I; LA—Left Atrium; LVEDD—Left Ventricular End-Diastolic Diameter; LDL—Low-Density Lipoprotein LVEF—Left Ventricular Ejection Fraction; LVESD—Left Ventricular End-Systolic Diameter; MAGGIC—Meta-Analysis Global Group in Chronic Heart Failure; NT-proBNP—N-terminal-pro hormone BNP; NYHA—New York Heart Association; PASP—Pulmonary Arterial Systolic Pressure; pVO_2_—Peak Oxygen Uptake; RER—Respiratory Exchange Ratio; SHFM—Seattle Heart Failure Model; TAPSE—Tricuspid Annular Plane Systolic Excursion; VE/VCO_2_—Ventilatory Inefficiency.

**Table 2 jcm-11-02935-t002:** Adverse events during 6 months follow-up.

	Number of Events
**Dapagliflozin + OMT**	
MACE Unplanned HF hospitalization All-cause death Heart transplant/LVAD implantation	3 2 0 0
**Control Group (maintain OMT)**	
MACE Unplanned HF hospitalization All-cause death Heart transplant/LVAD implantation	2 2 0 0

HF—Heart Failure; LVAD—Left Ventricular Assist Device; MACE—Major Adverse Cardiac Events; OMT—Optimal Medical Therapy.

**Table 3 jcm-11-02935-t003:** Comparison of clinical, echocardiographic and exercise parameters variation between groups.

					*p*-Value (Partial η^2^)		
	Dapagliflozin (*n* = 20)		Control (*n* = 20)		Time	Group	Interaction
	Baseline	6 m Fup	Baseline	6 m Fup			
NYHA Class III (No vs. Yes) (a)	6 (28.6%)	1 (4.8%)	2 (10.5%)	5 (26.3%)	**0.031 (8.00)**	0.087 (7.14)	**0.005 (0.041)**
Furosemide Dose (mg)	25.7 ± 29.8	20.9 ± 29.9	33.7 ± 32.6	38.9 ± 32.9	0.931 (0.00)	0.182 (0.05)	0.061 (0.09)
HFSS	8.7 ± 0.6	8.9 ± 0.8	8.6 ± 0.7	8.7 ± 0.9	0.087 (0.11)	0.512 (0.02)	0.639 (0.10)
SHFM	89.5 ± 6.7	92.5 ± 4.4	88.3 ± 4.9	87.6 ± 6.1	0.137 (0.06)	0.080 (0.09)	**0.014 (0.17)**
MAGGIC Score	18.1 ± 5.4	17.0 ± 5.0	17.8 ± 5.6	19.2 ± 7.4	0.992 (0.00)	0.550 (0.01)	0.073 (0.09)
Hemoglobin (g/dL)	14.0 ± 1.6	14.5 ± 1.6	13.8 ± 1.2	13.9 ± 1.6	0.091 (0.08)	0.422 (0.02)	0.238 (0.04)
Creatinine (mg/dL)	1.2 ± 0.3	1.2 ± 0.3	1.1 ± 0.3	1.2 ± 0.3	**0.033 (0.12)**	0.757 (0.00)	0.183 (0.05)
GFR (mL/min)	68.7 ± 23.8	66.8 ± 22.2	72.5 ± 17.1	65.9 ± 16.5	**0.010 (0.17)**	0.834 (0.00)	0.144 (0.06)
hs-cTnI (ng/mL)	9.2 ± 6.7	8.4 ± 5.2	21.9 ± 45.1	27.7 ± 43.4	0.935 (0.00)	0.161 (0.06)	0.200 (0.00)
NT-proBNP (pg/mL)	1201.5 ± 920.9	983.9 ± 823.5	1132.1 ± 1774.4	1782.4 ± 2513.7	0.162 (0.05)	0.485 (0.01)	**0.007 (0.19)**
HbA1c (%)	5.8 ± 0.4	5.6 ± 0.2	5.7 ± 0.5	6.1 ± 0.4	0.404 (0.02)	0.854 (0.00)	**<0.001 (0.28)**
LDL (mg/dL)	85.2 ± 33.2	86.9 ± 48.9	95.8 ± 36.9	88.8 ± 52.3	0.478 (0.01)	0.610 (0.01)	0.281 (0.03)
LVEDD (mm)	67.2 ± 8.4	63.0 ± 9.7	65.1 ± 6.0	65.8 ± 8.6	0.055 (0.10)	0.618 (0.01)	**0.010 (0.17)**
LVESD (mm)	49.4 ± 9.6	46.1 ± 13.4	46.9 ± 14.3	41.6 ± 10.8	**0.004 (0.23)**	0.760 (0.00)	0.460 (0.02)
LVEF (%)	33.5 ± 7.8	36.9 ± 8.5	34.5 ± 8.9	36.2 ± 8.1	0.057 (0.10)	0.759 (0.00)	0.512 (0.01)
GLS (%)	−8.2 ± 3.2	−10.4 ± 2.3	−8.4 ± 2.8	−6.8 ± 2.8	0.549 (0.02)	0.092 (0.11)	**<0.001 (0.39)**
TAPSE (mm)	19.0 ± 5.7	20.2 ± 5.7	18.6 ± 4.2	18.7 ± 4.2	0.376 (0.00)	0.883 (0.00)	0.376 (0.00)
LA Indexed Volume (ml/m^2^)	43.4 ± 12.2	36.5 ± 10.1	41.9 ± 17.5	48.1 ± 11.2	0.760 (0.00)	0.296 (0.03)	**<0.001 (0.08)**
PASP (mmHg)	36.6 ± 7.8	30.7 ± 8.6	29.8 ± 11.4	39.4 ± 10.4	0.560 (0.00)	0.344 (0.04)	**<0.001 (0.15)**
E/e’	13.3 ± 4.5	11.9 ± 3.8	11.8 ± 4.9	13.3 ± 7.5	0.938 (0.00)	0.783 (0.00)	**0.026 (0.16)**
CPET Duration (minutes)	10.0 ± 4.0	11.9 ± 4.6	10.9 ± 3.2	11.5 ± 4.4	**0.025 (0.17)**	0.755 (0.00)	0.225 (0.05)
Peak RER	1.10 ± 011	1.11 ± 0.12	1.09 ± 0.12	0.99 ± 0.09	**0.003 (0.27)**	0.185 (0.06)	**0.020 (0.18)**
pVO_2_ (ml/kg/min)	16.4 ± 3.9	19.5 ± 6.0	16.5 ± 5.1	16.6 ± 5.2	**0.018 (0.18)**	0.537 (0.01)	**0.027 (0.16)**
VE/VCO_2_	34.4 ± 7.9	33.6 ± 6.7	34.1 ± 8.9	37.4 ± 9.4	0.169 (0.06)	0.789 (0.00)	**0.027 (0.15)**
pVO_2_ at LANA	11.8 ± 2.8	12.4 ± 3.6	11.7 ± 3.4	12.9 ± 3.2	0.081 (0.11)	0.857 (0.00)	0.556 (0.01)
Cardiorrespiratory Optimal Point	30.7 ± 5.6	28.8 ± 4.5	28.0 ± 6.6	30.5 ± 8.2	0.719 (0.01)	0.476 (0.02)	**0.018 (0.21)**
HR at LANA	87.9 ± 14.8	91.0 ± 18.0	97.2 ± 21.6	94.3 ± 17.3	0.786 (0.00)	0.403 (0.03)	0.233 (0.05)
HRR1	21.2 ± 12.6	17.5 ± 13.2	21.0 ± 12.0	20.7 ± 10.8	0.369 (0.03)	0.943 (0.00)	0.583 (0.02)

Values are the mean ± SD, *n* (%), or median (interquartile range). Bold values denote statistical significance at the *p* < 0.05 level. CPET—Cardiopulmonary Exercise Testing; GLS—Global Longitudinal Strain; HbA1c—Glycated Hemoglobin; HFSS—Heart Failure Survival Score; hs-cTnI—High-Sensitivity Troponin I; LA—Left Atrium; LVEDD—Left Ventricular End-Diastolic Diameter; LDL—Low-Density Lipoprotein LVEF—Left Ventricular Ejection Fraction; LVESD—Left Ventricular End-Systolic Diameter; MAGGIC—Meta-Analysis Global Group in Chronic Heart Failure; NT-proBNP—N-terminal-pro hormone BNP; NYHA—New York Heart Association; PASP—Pulmonary Arterial Systolic Pressure; pVO_2_—Peak Oxygen Uptake; RER—Respiratory Exchange Ratio; SHFM—Seattle Heart Failure Model; TAPSE—Tricuspid Annular Plane Systolic Excursion; VE/VCO_2_—Ventilatory Inefficiency; 6 m Fup—6 months follow-up; (a) effect size calculated as odds ratio, reference category for NYHA = No.

## Data Availability

Data available on request due to restrictions, such as privacy.

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
