# Peer review of "Dapagliflozin Impact on the Exercise Capacity of Non-Diabetic Heart Failure with Reduced Ejection Fraction Patients"

_jcm, 2022, doi:10.3390/jcm11102935_

Round 1
Reviewer 1 Report
In this single center experience, Dapagliflozin use in non-diabetic HF patients with LVEF<50% proved to be a safe strategy and led to a significant improvement of natriuretic peptide levels, exercise performance and echocardiographic markers of LV systolic function at a 6-months follow-up.
My general impression is that the results are not surprising, but the study is still important. There are some drawbacks that have caught my attention.
Major concerns;
Because of the nature of study-design, the patients who could undergo CPET were enrolled, leading to selection bias. The inclusion/exclusion criteria should be described precisely.
Please show the number of non-diabetic HF patients screened in this institute during the enrollment period (e.g., the number of patients who were excluded in each exclusion criteria). These data is important to consider the generality of the results of this study.
I think that this randomized study data does not support the evidence of “real-word” patients. The patient characteristics of this study is different from those of HF registry database.
Baseline profile was different from that of the DAPA-HF trial (younger age in this study). (page 17 line 278-8)
Minor;
The difference in NT-pro BNP should be described as secondary endpoint in the Methods section.
The non-distributed data should be described as median [interquartile range].
LVEF < 40% is incorrect within Figure 1.
SHFM is incorrect within Results (SHFM-estimated X year survival rate?).
Reviewer 2 Report
Ferrerira Reis et al. studies the effect of Dapagliflozin in non-diabetic heart failure with reduced ejection fraction patients. The study shows some improvement in PvO2 after drug administration, suggesting that Dapagliflozin may positively impact the treatment of heart failure in this specific patient cohort. The writing style and English used in the manuscript are appropriate, but the analyses performed in the study present some limitations that question the author's conclusions.
Comments:
As the authors recognized in the “Limitations” section of the manuscript, the small number of patients used in the study may affect the validity and reproducibility of the work’s conclusions. To reproduce the study, the authors need to provide at least the typical deviation assumed during the obtention of the results.
The manuscript would benefit from adding a glossary of abbreviations and acronyms.
This reviewer misses in the manuscript some discussion about the possibility of using Dapagliflozin to treat HFpEF patients.
Instead of directly copying and pasting the graphs after performing the statistical analyses, the authors should take their time to improve the quality of the figures. This would help the reader to understand better the data shown. 1) The bars used to show statistical significance in Figures 3 and 4 shouldn’t touch the bar graphs. 2) The error bars in Figures 1, 3, and 4 shouldn’t interfere with the numeric value of the bar. 3) The numeric values of each variable should be easy to read. 4) In Figures 3 and 4, the name of the variables (X-axis) need to be located at the bottom of the graph to avoid overlap with the bar graphs.
Line 173. Instead of using a Wilcoxon test, this reviewer strongly recommends applying the GLM Repeated Measure. This model allows for comparing values along time, between groups, and the interaction between time and groups. In addition, the general linear model avoids increasing the test type I error (false positives), providing a better way to analyze the data.
Figure 3. The p-value of the “pVO2 at LANA” variable (0.552) does not match the value shown in table 4 (0.553).
To facilitate the table read, each significant p-value needs to be highlighted.
The variable “Diabetes” should be removed from tables 1 and 2 since it was used as an exclusion criterion.
Tables 1 and 2 should include a 0% in those variables with frequency = 0.
Table 3 misses the percentage values. The table also shows two 0 values that are tilted.
To avoid data duplication and facilitate the reading of the manuscript, tables 1 and 2 should be combined following this template:
Total (n=40) |
Dapagliflozin (n=20) |
Control (n=20) |
p value |
|
Age |
|
|
|
|
Male gender |
|
|
|
|
Ischemic etiology |
|
|
|
|
Previous MI |
|
|
|
|
Previous PCI |
|
|
|
|
Previous CABG |
|
|
|
|
Previous Valvular Heart Surgery |
|
|
|
|
HF hospitalizations in the previous 12 months |
|
|
|
|
Heart Failure Medication |
|
|
|
|
Beta-Blocker |
|
|
|
|
ACE-i/ ARB |
|
|
|
|
ARNI |
|
|
|
|
MRA |
|
|
|
|
Ivabradine |
|
|
|
|
Digitalis |
|
|
|
|
Oral Anticoagulation |
|
|
|
|
Furosemide Dose (mg) |
|
|
|
|
Ferric carboxymaltose |
|
|
|
|
Cardiac Rehabilitation |
|
|
|
|
Hypertension |
|
|
|
|
Dyslipidemia |
|
|
|
|
Diabetes |
|
|
|
|
Current or Former smoker |
|
|
|
|
Atrial Fibrillation |
|
|
|
|
ICD |
|
|
|
|
CRT Device |
|
|
|
|
Chronic Kidney Disease |
|
|
|
|
Peripheral Artery Disease |
|
|
|
|
COPD |
|
|
|
|
A similar modification should be done with table 4. See proposed template:
Dapagliflozin (N=20) |
Control (N=20) |
p_value for time |
p_value for group |
p_value for interaction |
|||||||
Baseline |
6-months follow-up |
Baseline |
6-months follow-up |
|
|||||||
Furosemide Dose (mg) |
|
|
|
|
|
|
|
|
|||
HFSS |
|
|
|
|
|
|
|
|
|||
… |
|
|
|
|
|
|
|
|
|||
Round 2
Reviewer 1 Report
Thank you for responding my suggestions.
Author Response
We are glad that you feel that all the questions have been clarified.

Reviewer 2 Report
In the new version of the manuscript, Ferreira et al. modified the tables and main text. However, there is still room for improvement, especially the table organization.
- Lines 175-178. The authors need to include the pooled standard deviation assumed to obtain the study’s sample size (16 patients).
- The second part of table 1 should be removed since this data already appear in table 3.
- Table 3 still misses the requested p-value for time, group and interaction.
- The new table 2, previously numbered table 3, still misses the percentage values, and the 0 values for “all-cause death” and “Heart transplant / LVAD implantation” variables are tilted.
- Table 1 misses the category for each represented variable. They added “Male” to the variable Gender, but this reviewer misses the other categories.
- The authors used bold numbers to highlight all the p-values, but this reviewer asked to highlight only the significant ones.
- Table 1. Some of the values of the “Heart Failure Medication” variable are tilted, and one of them (19 (95.0%)) is missing a bracket.
Author Response
We have addressed all the questions point-by-point and we feel that all the questions may have been clarified.
The pertinent changes have been included and marked in the text.
We do not include any further changes to the text due to space constraints. But if the editors find it useful, we will be available to do so.
Lines 175-178. The authors need to include the pooled standard deviation assumed to obtain the study’s sample size (16 patients).
The assumed pooled standard deviation used to obtain the study’s sample size was 2 ml/kg/min.
The second part of table 1 should be removed since this data already appear in table 3.
The second part of table 1 was removed as requested.
Table 3 still misses the requested p-value for time, group and interaction.
The aforementioned p values were added as requested by Reviewer 2. Repeated measures ANOVA with time as within subjects effect and treatment with dapagliflozin as between subjects effect were implemented to assess the effect of time (baseline vs 6 months), treatment (yes/ no) and their interaction regarding con-tinuous outcomes. Partial eta2 was calculated for assessing effect size considering >0.2 (small), >0.5 (moderate) and >0.8 (large). Generalized estimating equations were im-plemented to assess NYHA III class (no vs yes) change across time, groups of treatment with dapagliflozin and time x treatment interaction. Effect size was assessed with odds ratios.
The new table 2, previously numbered table 3, still misses the percentage values, and the 0 values for “all-cause death” and “Heart transplant / LVAD implantation” variables are tilted.
The percentage values were added and the table was formatted and corrected as requested.
Table 1 misses the category for each represented variable. They added “Male” to the variable Gender, but this reviewer misses the other categories.
The authors added the respective missing category for each represented variable as requested.
The authors used bold numbers to highlight all the p-values, but this reviewer asked to highlight only the significant ones.
Authors have formatted all tables, and now only statistically significant p-values are highlighted in bold.
Table 1. Some of the values of the “Heart Failure Medication” variable are tilted, and one of them (19 (95.0%)) is missing a bracket.
The table was formatted and corrected as requested.
